# Fast Low-rank Metric Learning for Large-scale and High-dimensional Data

**Han Liu** [†], **Zhizhong Han**[‡], **Yu-Shen Liu**[† *], **Ming Gu**[†]

† School of Software, Tsinghua University, Beijing, China
BNRist & KLISS, Beijing, China
‡ Department of Computer Science, University of Maryland, College Park, USA
liuhan15@mails.tsinghua.edu.cn    h312h@umd.edu
liuyushen@tsinghua.edu.cn    guming@tsinghua.edu.cn

## Abstract

Low-rank metric learning aims to learn better discrimination of data subject to low-rank constraints. It keeps the intrinsic low-rank structure of datasets and reduces the time cost and memory usage in metric learning. However, it is still a challenge for current methods to handle datasets with both high dimensions and large numbers of samples. To address this issue, we present a novel fast low-rank metric learning (FLRML) method. FLRML casts the low-rank metric learning problem into an unconstrained optimization on the Stiefel manifold, which can be efficiently solved by searching along the descent curves of the manifold. FLRML significantly reduces the complexity and memory usage in optimization, which makes the method scalable to both high dimensions and large numbers of samples. Furthermore, we introduce a mini-batch version of FLRML to make the method scalable to larger datasets which are hard to be loaded and decomposed in limited memory. The outperforming experimental results show that our method is with high accuracy and much faster than the state-of-the-art methods under several benchmarks with large numbers of high-dimensional data. Code has been made available at https://github.com/highan911/FLRML.

## 1   Introduction

Metric learning aims to learn a distance (or similarity) metric from supervised or semi-supervised information, which provides better discrimination between samples. Metric learning has been widely used in various area, such as dimensionality reduction [1, 2, 3], robust feature extraction [4, 5] and information retrieval [6, 7]. For existing metric learning methods, the huge time cost and memory usage are major challenges when dealing with high-dimensional datasets with large numbers of samples. To resolve this issue, low-rank metric learning (LRML) methods optimize a metric matrix subject to low-rank constraints. These methods tend to keep the intrinsic low-rank structure of the dataset, and also, reduce the time cost and memory usage in the learning process. Reducing the matrix size in optimization is an important idea to reduce time and memory usage. However, the size of the matrix to be optimized still increases linearly or squarely with either the dimensions, the number of samples, or the number of pairwise/triplet constraints. As a result, it is still a research challenge when dealing with the metric learning task on datasets with both high dimensions and large numbers of samples [8, 9].

To address this issue, we present a Fast Low-Rank Metric Learning (FLRML). In contrast to state-of-the-art methods, FLRML introduces a novel formulation to better employ the low rank constraints to further reduce the complexity, the size of involved matrices, which enables FLRML to achieve high

accuracy and faster speed on large numbers of high-dimensional data. Our main contributions are listed as follows.

- Modeling the constrained metric learning problem as an unconstrained optimization that can be efficiently solved on the Stiefel manifold, which makes our method scalable to large numbers of samples and constraints.

- Reducing the matrix size and complexity in optimization as much as possible while ensuring the accuracy, which makes our method scalable to both large numbers of samples and high dimensions.

- Furthermore, a mini-batch version of FLRML is proposed to make the method scalable to larger datasets which are hard to be fully loaded in memory.

## 2    Related Work

In metric learning tasks, the training dataset can be represented as a matrix $\mathbf{X} = [\mathbf{x}_1, ..., \mathbf{x}_n] \in \mathbb{R}^{D \times n}$, where $n$ is the number of training samples and each sample $\mathbf{x}_i$ is with $D$ dimensions. Metric learning methods aim to learn a metric matrix $\mathbf{M} \in \mathbb{R}^{D \times D}$ from the training set in order to obtain better discrimination between samples. Some low-rank metric learning (LRML) methods have been proposed to obtain the robust metric of data, and to reduce the computational costs for high-dimensional metric learning tasks. Since the optimization with fixed low-rank constraint is nonconvex, the naive gradient descent methods are easy to fall into bad local optimal solutions [2, 10]. In terms of different strategies to remedy this issue, the existing LRML methods can be roughly divided into the following two categories.

One type of method [1, 11, 12, 13, 14] introduces the low-rankness encouraging norms (such as nuclear norm) as regularization, which relaxes the nonconvex low-rank constrained problems to convex problems. The two disadvantages of such methods are: (1) the norm regularization can only encourage the low-rankness, but cannot limit the upper bound of rank; (2) the matrix to be optimized is still the size of either $D^2$ or $n^2$.

Another type of method [2, 3, 15, 16, 17, 18] considers the low-rank constrained space as Riemannian manifold. This type of method can obtain high-quality solutions of the nonconvex low-rank constrained problems. However, for these methods, the matrices to be optimized are at least a linear size of either $D$ or $n$. The performance of these methods is still suffering on large-scale and high-dimensional datasets.

Besides low-rank metric learning methods, there are some other types of methods for speeding up metric learning on large and high-dimensional datasets. Online metric leaning [6, 7, 19, 20] randomly takes one sample at each time. Sparse metric leaning [21, 22, 23, 24, 25, 26] represents the metric matrix as the sparse combination of some pre-generated rank-1 bases. Non-iterative metric leaning [27, 28] avoids iterative calculation by providing explicit optimal solutions. In the experiments, some state-of-the-art methods of these types will also be included for comparison. Compared with these methods, our method also has advantage in time and memory usage on large-scale and high-dimensional datasets. A literature review of many available metric learning methods is beyond the scope of this paper. The reader may consult Refs. [8, 9, 29] for detailed expositions.

## 3    Fast Low-Rank Metric Learning (FLRML)

The metric matrix $\mathbf{M}$ is usually semidefinite, which guarantees the non-negative distances and non-negative self-similarities. A semidefinite $\mathbf{M}$ can be represented as the transpose multiplication of two identical matrices, $\mathbf{M} = \mathbf{L}^\top \mathbf{L}$, where $\mathbf{L} \in \mathbb{R}^{d \times D}$ is a row-full-rank matrix, and $\mathrm{rank}(\mathbf{M}) = d$. Using the matrix $\mathbf{L}$ as a linear transformation, the training set $\mathbf{X}$ can be mapped into $\mathbf{Y} \in \mathbb{R}^{d \times n}$, which is denoted by $\mathbf{Y} = \mathbf{L}\mathbf{X}$. Each column vector $\mathbf{y}_i$ in $\mathbf{Y}$ is the corresponding $d$-dimensional vector of the column vector $\mathbf{x}_i$ in $\mathbf{X}$.

In this paper, we present a fast low-rank metric learning (FLRML) method, which typically learns the low-rank cosine similarity metric from triplet constraints $\mathcal{T}$. The cosine similarity between a pair of vectors $(\mathbf{x}_i, \mathbf{x}_j)$ is measured by their corresponding low dimensional vector $(\mathbf{y}_i, \mathbf{y}_j)$ as $sim(\mathbf{x}_i, \mathbf{x}_j) = \frac{\mathbf{y}_i^\top \mathbf{y}_j}{||\mathbf{y}_i|| \, ||\mathbf{y}_j||}$. Each constraint $\{i, j, k\} \in \mathcal{T}$ refers to the comparison of a pair of similarities $sim(\mathbf{x}_i, \mathbf{x}_j) > sim(\mathbf{x}_i, \mathbf{x}_k)$.

To solve the problem of metric learning on large-scale and high-dimensional datasets, our motivation is to reduce matrix size and complexity in optimization as much as possible while ensuring the accuracy. To address this issue, our idea is to embed the triplet constraints into a matrix $\mathbf{K}$, so that the constrained metric learning problem can be casted into an unconstrained optimization in the "form" of $\text{tr}(\mathbf{WK})$, where $\mathbf{W}$ is a low-rank semidefinite matrix to be optimized (Section 3.1). By reducing the sizes of $\mathbf{W}$ and $\mathbf{K}$ to $\mathbb{R}^{r \times r}$ ($r = \text{rank}(\mathbf{X})$), the complexity and memory usage are greatly reduced. An unconstrained optimization in this form can be efficiently solved on the Stiefel manifold (Section 3.2). In addition, a mini-batch version of FLRML is proposed, which makes our method scalable to larger datasets that are hard to be fully loaded and decomposed in the memory (Section 3.3).

## 3.1 Forming the Objective Function

Using margin loss, each triplet $t = \{i, j, k\}$ in $\mathcal{T}$ corresponds to a loss function $l(\{i, j, k\}) = \max(0, m - sim(\mathbf{x}_i, \mathbf{x}_j) + sim(\mathbf{x}_i, \mathbf{x}_k))$. A naive idea is to sum the loss functions, but when $n$ and $|\mathcal{T}|$ are very large, the evaluation of loss functions will be time consuming. Our idea is to embed the evaluation of loss functions into matrices to speed up their calculation.

For each triplet $t = \{i, j, k\}$ in $\mathcal{T}$, a matrix $\mathbf{C}^{(t)}$ with the size $n \times n$ is generated, which is a sparse matrix with $c_{ji}^{(t)} = 1$ and $c_{ki}^{(t)} = -1$. The summation of all $\mathbf{C}^{(t)}$ matrices is represented as $\mathbf{C} = \sum_{t \in \mathcal{T}} \mathbf{C}^{(t)}$. The matrix $\mathbf{YC}$ is with the size $\mathbb{R}^{d \times n}$. Let $\mathcal{T}_i$ be the subset of triplets with $i$ as the first item, and $\tilde{\mathbf{y}}_i$ be the $i$-th column of $\mathbf{YC}$, then $\tilde{\mathbf{y}}_i$ can be written as $\tilde{\mathbf{y}}_i = \sum_{\{i,j,k\} \in \mathcal{T}_i} (-\mathbf{y}_j + \mathbf{y}_k)$. This is the sum of negative samples minus the sum of positive samples for the set $\mathcal{T}_i$. By multiplying on $\frac{1}{|\mathcal{T}_i|+1}$ on both sides, we can get $\frac{1}{|\mathcal{T}_i|+1}\tilde{\mathbf{y}}_i$ as the mean of negative samples minus the mean of positive samples (in which "$|\mathcal{T}_i| + 1$" is to avoid zero on the denominator).

Let $z_i = \frac{1}{|\mathcal{T}_i|+1}\mathbf{y}_i^\top \tilde{\mathbf{y}}_i$, then by minimizing $z_i$, the vector $\mathbf{y}_i$ tends to be closer to the positive samples than the negative samples. Let $\mathbf{T}$ be a diagonal matrix with $T_{ii} = \frac{1}{|\mathcal{T}_i|+1}$, then $z_i$ is the $i$-th diagonal element of $-\mathbf{Y}^\top \mathbf{YCT}$. The loss function can be constructed by putting $z_i$ into the margin loss as $L(\mathcal{T}) = \sum_{i=1}^n \max(0, z_i + m)$. A binary function $\lambda(x)$ is defined as: if $x > 0$, then $\lambda(x) = 1$; otherwise, $\lambda(x) = 0$. By introducing the function $\lambda(x)$, the loss function $L(\mathcal{T})$ can be written as $L(\mathcal{T}) = \sum_{i=1}^n (z_i + m)\lambda(z_i + m)$, which can be further represented in matrices:

$$L(\mathcal{T}) = -\text{tr}(\mathbf{Y}^\top \mathbf{YCT\Lambda}) + M(\mathbf{\Lambda}), \tag{1}$$

where $\mathbf{\Lambda}$ is a diagonal matrix with $\Lambda_{ii} = \lambda(z_i + m)$, and $M(\mathbf{\Lambda}) = m \sum_{i=1}^n \Lambda_{ii}$ is the sum of constant terms in the margin loss.

In Eq.(1), $\mathbf{Y} \in \mathbb{R}^{d \times n}$ is the optimization variable. For any value of $\mathbf{Y}$, the corresponding value of $\mathbf{L}$ can be obtained by solving the linear equation group $\mathbf{Y} = \mathbf{LX}$. A minimum norm least squares solution of $\mathbf{Y} = \mathbf{LX}$ is $\mathbf{L} = \mathbf{YV\Sigma^{-1}U^\top}$, where $[\mathbf{U} \in \mathbb{R}^{D \times r}, \mathbf{\Sigma} \in \mathbb{R}^{r \times r}, \mathbf{V} \in \mathbb{R}^{n \times r}]$ is the SVD of $\mathbf{X}$. Based on this, the size of the optimization variable can be reduced from $\mathbb{R}^{d \times n}$ to $\mathbb{R}^{d \times r}$, as shown in Theorem 1.

**Theorem 1.** $\{\mathbf{Y} : \mathbf{Y} = \mathbf{BV}^\top, \mathbf{B} \in \mathbb{R}^{d \times r}\}$ *is a subset of* $\mathbf{Y}$ *that covers all the possible minimum norm least squares solutions of* $\mathbf{L}$.

*Proof.* By substituting $\mathbf{Y} = \mathbf{BV}^\top$ into $\mathbf{L} = \mathbf{YV\Sigma^{-1}U^\top}$,

$$\mathbf{L} = \mathbf{B\Sigma^{-1}U^\top}. \tag{2}$$

Since $\mathbf{U}$ and $\mathbf{\Sigma}$ are constants, then $\mathbf{B} \in \mathbb{R}^{d \times r}$ covers all the possible minimum norm least squares solutions of $\mathbf{L}$. □

By substituting $\mathbf{Y} = \mathbf{BV}^\top$ into Eq.(1), the sizes of $\mathbf{W}$ and $\mathbf{K}$ can be reduced to $\mathbb{R}^{r \times r}$, which are represented as

$$L(\mathcal{T}) = -\text{tr}(\mathbf{B}^\top \mathbf{BV}^\top \mathbf{CT\Lambda V}) + M(\mathbf{\Lambda}). \tag{3}$$

This function is in the form of $\text{tr}(\mathbf{WK})$, where $\mathbf{W} = \mathbf{B}^\top \mathbf{B}$ and $\mathbf{K} = -\mathbf{V}^\top \mathbf{CT\Lambda V}$. The size of $\mathbf{W}$ and $\mathbf{K}$ are reduced to $\mathbb{R}^{r \times r}$, and $r \leq \min(n, D)$. In addition, $\mathbf{V}^\top \mathbf{CT} \in \mathbb{R}^{r \times n}$ is a constant matrix that will not change in the process of optimization. So this model is with low complexity.

It should be noted that the purpose of SVD here is not for approximation. If all the ranks of $\mathbf{X}$ are kept, i.e. $r = \text{rank}(\mathbf{X})$, the solutions are supposed to be exact. In practice, it is also reasonable to neglect the smallest eigenvalues of $\mathbf{X}$ to speed up the calculation. In the experiments, an upper bound is set as $r = \min(\text{rank}(\mathbf{X}), 3000)$, since most computers can easily handle a matrix of $3000^2$ size, and the information in most of the datasets can be preserved well.

## 3.2 Optimizing on the Stiefel Manifold

The matrix $\mathbf{W} = \mathbf{B}^\top \mathbf{B}$ is the low-rank semidefinite matrix to be optimized. Due to the non-convexity of low-rank semidefinite optimization, directly optimizing $\mathbf{B}$ in the linear space often falls into bad local optimal solutions [2, 10]. The mainstream strategy of low-rank semidefinite problem is to achieve the optimization on manifolds.

The Stiefel manifold $\text{St}(d, r)$ is defined as the set of $r \times d$ column-orthogonal matrices, i.e., $\text{St}(d, r) = \{\mathbf{P} \in \mathbb{R}^{r \times d} : \mathbf{P}^\top \mathbf{P} = \mathbf{I}_d\}$. Any semidefinite $\mathbf{W}$ with $\text{rank}(\mathbf{W}) = d$ can be represented as $\mathbf{W} = \mathbf{P S P}^\top$, where $\mathbf{P} \in \text{St}(d, r)$ and $\mathbf{S} \in \{\text{diag}(\mathbf{s}) : \mathbf{s} \in \mathbb{R}_+^d\}$. Since $\mathbf{P}$ is already restricted on the Stiefel manifold, we only need to add regularization term for $\mathbf{s}$. We want to guarantee the existence of dense finite optimal solution of $\mathbf{s}$, so the L2-norm of $\mathbf{s}$ is used as a regularization term. By adding $\frac{1}{2}||\mathbf{s}||^2$ into Eq.(3), we get

$$f_0(\mathbf{W}) = \text{tr}(\mathbf{P S P}^\top \mathbf{K}) + \frac{1}{2}||\mathbf{s}||^2 + M(\mathbf{\Lambda}) = \sum_{i=1}^d (\frac{1}{2}s_i^2 + s_i \mathbf{p}_i^\top \mathbf{K} \mathbf{p}_i) + M(\mathbf{\Lambda}), \qquad (4)$$

where $\mathbf{p}_i$ is the $i$-th column of $\mathbf{P}$.

Let $k_i = -\mathbf{p}_i^\top \mathbf{K} \mathbf{p}_i$. Since $f_0(\mathbf{W})$ is a quadratic function for each $s_i$, for any value of $\mathbf{P}$, the only corresponding optimal solution of $\mathbf{s} \in \mathbb{R}_+^d$ is

$$\hat{\mathbf{s}} = \{[\hat{s}_1, ..., \hat{s}_d]^\top : \hat{s}_i = \max(0, k_i)\}. \qquad (5)$$

By substituting the $\hat{s}_i$ values into Eq.(4), the existence of $\mathbf{s}$ in $f_0(\mathbf{W})$ can be eliminated, which converts it to a new function $f(\mathbf{P})$ that is only relevant with $\mathbf{P}$, as shown in the following Theorem 2.

**Theorem 2.**

$$f(\mathbf{P}) = -\frac{1}{2} \sum_{i=1}^d (\max(0, k_i) k_i) + M(\mathbf{\Lambda}). \qquad (6)$$

*Proof.* An original form of $f(\mathbf{P})$ can be obtained by substituting Eq.(5) into Eq.(4):
$\sum_{i=1}^d ((\frac{1}{2}\max(0, k_i) - k_i)\max(0, k_i)) + M(\mathbf{\Lambda})$.
The $f(\mathbf{P})$ in Eq.(6) is equal to this formula in both $k_i \leq 0$ and $k_i > 0$ conditions. $\qquad \square$

In order to make the gradient of $f(\mathbf{P})$ continuous, and to keep $\mathbf{s}$ dense and positive, we adopt function $\mu(x) = -\log(\sigma(-x))$ as the smoothing of $\max(0, x)$ [2], where $\sigma(x) = 1/(1 + \exp(-x))$ is the sigmoid function. Function $\mu(x)$ satisfies $\lim_{x \to +\infty} = x$ and $\lim_{x \to -\infty} = 0$. The derivative of $\mu(x)$ is $d\mu(x)/dx = \sigma(x)$. Figure 1 displays the sample plot of $\max(0, x)$ and $\mu(x)$. Using this smoothed

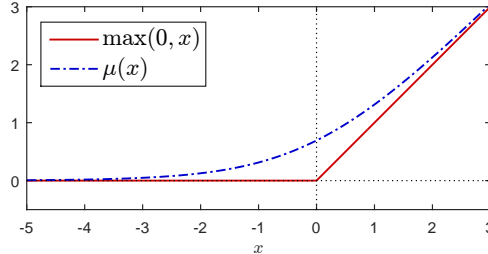

Figure 1: A sample plot of $\max(0, x)$ and $\mu(x)$.

| **Algorithm 1** FLRML | **Algorithm 2** M-FLRML |
|---|---|
| 1: **Input:** the data matrix $\mathbf{X} \in \mathbb{R}^{D \times n}$, the sparse supervision matrix $\mathbf{C} \in \mathbb{R}^{n \times n}$, the low-rank constraint $d$ <br> 2: $[\mathbf{U} \in \mathbb{R}^{D \times r}, \mathbf{\Sigma} \in \mathbb{R}^{r \times r}, \mathbf{V} \in \mathbb{R}^{n \times r}] \leftarrow \text{SVD}(\mathbf{X})$ <br> 3: Calculating constant matrix $\mathbf{V}^\top \mathbf{C} \mathbf{T}$ <br> 4: Randomly initialize $\mathbf{P} \in \text{St}(d, r)$ <br> 5: Initialize $\mathbf{S}$ satisfies Eq.(8) <br> 6: **repeat** <br> 7:     Update $\mathbf{\Lambda}$ and $\mathbf{K}$ by Eq.(3) <br> 8:     Update $\mathbf{G}$ by Eq.(9) <br> 9:     Update $\mathbf{P}$ and $\mathbf{S}$ by searching on $h(\tau)$ in Eq.(10) <br> 10: **until** convergence <br> 11: **Output:** $\mathbf{L} \leftarrow \sqrt{\mathbf{S}} \mathbf{P}^\top \mathbf{\Sigma}^{-1} \mathbf{U}^\top$ | 1: **Input:** the data matrices $\mathbf{X}_I \in \mathbb{R}^{D \times n_I}$, the supervision matrices $\mathbf{C}_I \in \mathbb{R}^{n_I \times n_I}$, the low-rank constraint $d$ <br> 2: Randomly initialize $\mathbf{L} \in \mathbb{R}^{d \times D}$ <br> 3: **for** $I = 1 : T$ **do** <br> 4:     $[\mathbf{U}_I, \mathbf{\Sigma}_I, \mathbf{V}_I] \leftarrow \text{SVD}(\mathbf{X}_I)$ <br> 5:     Get $\mathbf{P}_I$ and $\mathbf{s}_I$ by Eq.(11) <br> 6:     Update $\mathbf{\Lambda}$ and $\mathbf{K}$ by Eq.(3) <br> 7:     Update $\mathbf{P}_I$ and $\mathbf{s}_I$ by Eq.(12) and Eq.(5) <br> 8:     Get $\Delta \mathbf{L}$ by Eq.(13) <br> 9:     $\mathbf{L} \leftarrow \mathbf{L} + \frac{1}{\sqrt{I}} \Delta \mathbf{L}$ <br> 10: **end for** <br> 11: **Output:** $\mathbf{L}$ |

function, the loss function $f(\mathbf{P})$ is redefined as

$$f(\mathbf{P}) = -\frac{1}{2} \sum_{i=1}^{d} (\mu(k_i) k_i) + M(\mathbf{\Lambda}). \tag{7}$$

The initialization of $\mathbf{S} \in \{\text{diag}(\mathbf{s}) : \mathbf{s} \in \mathbb{R}_+^d\}$ needs to satisfy the condition

$$\mathbf{s} = \hat{\mathbf{s}}. \tag{8}$$

When $\mathbf{P}$ is fixed, $\hat{\mathbf{s}}$ is a linear function of $\mathbf{K}$ (see Eq.(5)), $\mathbf{K}$ is a linear function of $\mathbf{\Lambda}$ (see Eq.(3)), and the 0-1 values in $\mathbf{\Lambda}$ are relevant with $\mathbf{s}$ (see Eq.(3)). So Eq.(8) is a nonlinear equation in a form that can be easily solved iteratively by updating $\mathbf{s}$ with $\hat{\mathbf{s}}$. Since $\hat{\mathbf{s}} \in \mathcal{O}(\mathbf{K}^1)$, $\mathbf{K} \in \mathcal{O}(\mathbf{\Lambda}^1)$, and $\mathbf{\Lambda} \in \mathcal{O}(\mathbf{s}^0)$, this iterative process has a superlinear convergence rate.

To solve the model of this paper, for a matrix $\mathbf{P} \in \text{St}(d, r)$, we need to get its gradient $\mathbf{G} = \frac{\partial f(\mathbf{P})}{\partial \mathbf{P}}$.
**Theorem 3.**

$$\mathbf{G} = \frac{\partial f(\mathbf{P})}{\partial \mathbf{P}} = -(\mathbf{K} + \mathbf{K}^\top) \mathbf{P} \text{diag}(\mathbf{q}), \tag{9}$$

where $q_i = -\frac{1}{2}(\mu(k_i) + k_i \sigma(k_i))$.

*Proof.* Since $q_i = \frac{\partial f(\mathbf{P})}{\partial k_i} = -\frac{1}{2}(\mu(k_i) + k_i \sigma(k_i))$, the gradient can be derived from $\frac{\partial f(\mathbf{P})}{\partial \mathbf{P}} = \sum_{i=1}^{d} q_i \frac{\partial k_i}{\partial \mathbf{P}}$.
The $\frac{\partial k_i}{\partial \mathbf{P}}$ can be easily obtained since $k_i = -\mathbf{p}_i^\top \mathbf{K} \mathbf{p}_i$. $\qquad\qquad\square$

For solving optimizations on manifolds, the commonly used method is the "projection and retraction", which first projects the gradient $\mathbf{G}$ onto the tangent space of the manifold as $\hat{\mathbf{G}}$, and then retracts $(\mathbf{P} - \hat{\mathbf{G}})$ back to the manifold. For Stiefel manifold, the projection of $\mathbf{G}$ on the tangent space is $\hat{\mathbf{G}} = \mathbf{G} - \mathbf{P} \mathbf{G}^\top \mathbf{P}$ [10]. The retraction of the matrix $(\mathbf{P} - \hat{\mathbf{G}})$ to the Stiefel manifold can be represented as $\text{retract}(\mathbf{P} - \hat{\mathbf{G}})$, which is obtained by setting all the singular values of $(\mathbf{P} - \hat{\mathbf{G}})$ to 1 [30].

For Stiefel manifolds, we adopt a more efficient algorithm [10], which performs a non-monotonic line search with Barzilai-Borwein step length [31, 32] on a descent curve of the Stiefel manifold. A descent curve with parameter $\tau$ is defined as

$$h(\tau) = (\mathbf{I} + \frac{\tau}{2} \mathbf{H})^{-1} (\mathbf{I} - \frac{\tau}{2} \mathbf{H}) \mathbf{P}, \tag{10}$$

where $\mathbf{H} = \mathbf{G} \, \mathbf{P}^\top - \mathbf{P} \, \mathbf{G}^\top$. The optimization is performed by searching the optimal $\tau$ along the descent curve. The Barzilai-Borwein method predicts a step length according to the step lengths in previous iterations, which makes the method converges faster than the "projection and retraction".

The outline of the FLRML algorithm is shown in **Algorithm 1**. It can be mainly divided into four stages: SVD preprocessing (line 2), constant initializing (line 3), variable initializing (lines 4 and 5), and the iterative optimization (lines 6 to 11). In one iteration, the complexity of each step is: **(a)** updating $\mathbf{Y}$ and $\mathbf{\Lambda}$: $\mathcal{O}(nrd)$; **(b)** updating $\mathbf{K}$: $\mathcal{O}(nr^2)$; **(c)** updating $\mathbf{G} : \mathcal{O}(r^2 d)$; **(d)** optimizing $\mathbf{P}$ and $\mathbf{S}$: $\mathcal{O}(rd^2)$.

### 3.3 Mini-batch FLRML

In FLRML, the maximum size of constant matrices in the iterations is only $\mathbb{R}^{r \times n}$ ($\mathbf{V}$ and $\mathbf{V}^\top \mathbf{CT}$), and the maximum size of variable matrices is only $\mathbb{R}^{r \times r}$. Smaller matrix size theoretically means the ability to process larger datasets on the same size of memory. However, in practice, we find that the bottleneck is not the optimization process of FLRML. On large-scale and high-dimensional datasets, SVD preprocessing may take more time and memory than the FLRML optimization process. And for very large datasets, it will be difficult to load all data into memory. In order to break the bottleneck, and make our method scalable to larger numbers of high-dimensional data in limited memory, we further propose Mini-batch FLRML (M-FLRML).

Inspired by the stochastic gradient descent method [18, 33], M-FLRML calculates a descent direction from each mini-batch of data, and updates $\mathbf{L}$ at a decreasing ratio. For the $I$-th mini-batch, we randomly select $N_t$ triplets from the triplet set, and use the union of the samples to form a mini-batch with $n_I$ samples. Considering that the Stiefel manifold $\mathrm{St}(d, r)$ requires $r \geq d$, if the number of samples in the union of triplets is less than $d$, we randomly add some other samples to make $n_I > d$. The matrix $\mathbf{X}_I \in \mathbb{R}^{D \times n_I}$ is composed of the extracted columns from $\mathbf{X}$, and $\mathbf{C}_I \in \mathbb{R}^{n_I \times n_I}$ is composed of the corresponding columns and rows in $\mathbf{C}$.

The objective $f_0(\mathbf{W})$ in Eq.(4) consists of small matrices with size $\mathbb{R}^{r \times r}$ and $\mathbb{R}^{r \times d}$. Our idea is to first find the descent direction for small matrices, and then maps it back to get the descent direction of large matrix $\mathbf{L} \in \mathbb{R}^{d \times D}$. Matrix $\mathbf{X}_I$ can be decomposed as $\mathbf{X}_I = \mathbf{U}_I \mathbf{\Sigma}_I \mathbf{V}_I^\top$, and the complexity of decomposition is significantly reduced from $\mathcal{O}(Dnr)$ to $\mathcal{O}(Dn_I^2)$ on this mini-batch. According to Eq.(2), a matrix $\mathbf{B}_I$ can be represented as $\mathbf{B}_I = \mathbf{L}\mathbf{U}_I\mathbf{\Sigma}_I$. Using SVD, matrix $\mathbf{B}_I$ can be decomposed as $\mathbf{B}_I = \mathbf{Q}_I \mathrm{diag}(\sqrt{\mathbf{s}_I})\mathbf{P}_I^\top$, and then the variable $\mathbf{W}$ for objective $f_0(\mathbf{W})$ can be represented as

$$\mathbf{W} = \mathbf{B}_I^\top \mathbf{B}_I = \mathbf{P}_I \mathrm{diag}(\mathbf{s}_I)\mathbf{P}_I^\top. \tag{11}$$

In FLRML, in order to convert $f_0(\mathbf{W})$ into $f(\mathbf{P})$, the initial value of $\mathbf{s}$ satisfies the condition $\mathbf{s} = \hat{\mathbf{s}}$. But in M-FLRML, $\mathbf{s}_I$ is generated from $\mathbf{B}_I$, so generally this condition is not satisfied. So instead, we take $\mathbf{P}_I$ and $\mathbf{s}_I$ as two variables, and find the descent direction of them separately. In Mini-batch FLRML, when a different mini-batch is taken in next iteration, the predicted Barzilai-Borwein step length tends to be improper, so we use "projection and retraction" instead. The updated matrix $\hat{\mathbf{P}}_I$ is obtained as

$$\hat{\mathbf{P}}_I = \mathrm{retract}(\mathbf{P}_I - \mathbf{G}_I + \mathbf{P}_I \mathbf{G}_I^\top \mathbf{P}_I). \tag{12}$$

For $\mathbf{s}_I$, we use Eq.(5) to get an updated vector $\hat{\mathbf{s}}_I$. Then the updated matrix for $\mathbf{B}_I$ can be obtained as $\hat{\mathbf{B}}_I = \mathbf{Q}_I \mathrm{diag}(\sqrt{\hat{\mathbf{s}}_I})\hat{\mathbf{P}}_I^\top$. By mapping $\hat{\mathbf{B}}_I$ back to the high-dimensional space, the descent direction of $\mathbf{L}$ can be obtained as

$$\Delta \mathbf{L} = \mathbf{Q}_I \mathrm{diag}(\sqrt{\hat{\mathbf{s}}_I})\hat{\mathbf{P}}_I^\top \mathbf{\Sigma}_I^{-1} \mathbf{U}_I^\top - \mathbf{L}. \tag{13}$$

For the $I$-th mini-batch, $\mathbf{L}$ is updated at a decreasing ratio as $\mathbf{L} \leftarrow \mathbf{L} + \frac{1}{\sqrt{I}}\Delta \mathbf{L}$. The theoretical analysis of the stochastic strategy which updates in step sizes by $\frac{1}{\sqrt{I}}$ can refer to the reference [18]. The outline of M-FLRML is shown in **Algorithm 2**.

## 4 Experiments

### 4.1 Experiment Setup

In the experiments, our **FLRML** and **M-FLRML** are compared with 5 state-of-the-art low-rank metric learning methods, including **LRSML** [1], **FRML** [2], **LMNN** [34], **SGDINC** [18], and **DRML** [3]. For these methods, the complexities, maximum variable size and maximum constant size in one iteration are compared in Table 2. Considering that $d \ll D$ and $n_I \ll n$, the relatively small items in the table are omitted.

In addition, four state-of-the-art metric learning methods of other types are also compared, including one sparse method (**SCML** [23]), one online method (**OASIS** [6]), and two non-iterative methods (**KISSME** [27], **RMML** [28]).

The methods are evaluated on eight datasets with high dimensions or large numbers of samples: three datasets **NG20**, **RCV1-4** and **TDT2-30** derived from three text collections respectively [35, 36]; one

Table 1: The datasets used in the experiments.

| dataset | $D$ | $n$ | $n_{test}$ | $n_{cat}$ |
|---|---|---|---|---|
| NG20 | 62,061 | 15,935 | 3,993 | 20 |
| RCV1 | 29,992 | 4,813 | 4,812 | 4 |
| TDT2 | 36,771 | 4,697 | 4,697 | 30 |
| MNIST | 780 | 60,000 | 10,000 | 10 |
| M10-16 | 4,096 | 47,892 | 10,896 | 10 |
| M40-16 | 4,096 | 118,116 | 29,616 | 40 |
| M10-100 | 1,000,000 | 47,892 | 10,896 | 10 |
| M40-100 | 1,000,000 | 118,116 | 29,616 | 40 |

Table 2: The complexity, variable matrix size and constant matrix size of 7 low-rank metric learning methods (in one iteration).

| methods | complexity | size(var) | size(const) |
|---|---|---|---|
| LMNN | $|\mathcal{T}|r^2 + r^3$ | $|\mathcal{T}|r + r^2$ | $nr$ |
| LRSML | $n^2 d$ | $n^2$ | $n^2$ |
| FRML | $Dn^2 + D^2 d$ | $D^2$ | $Dn$ |
| DRML | $D^2|\mathcal{T}| + D^2 d$ | $Dd$ | $D|\mathcal{T}|$ |
| SGDINC | $Dd^2 + Dn_I^2$ | $Dd$ | $Dn_I$ |
| FLRML | $nrd + nr^2$ | $r^2 + nd$ | $nr$ |
| M-FLRML | $Dn_I^2 + Dn_I d$ | $Dd$ | $Dn_I$ |

Table 3: The classification accuracy (left) and training time (right, in seconds) of 7 metric learning methods with SVD preprocessing.

| datasets | NG20 | | RCV1 | | TDT2 | | MNIST | | M10-16 | | M40-16 | |
|---|---|---|---|---|---|---|---|---|---|---|---|---|
| $t_{svd}$ | | 191 | | 91 | | 107 | | 10 | | 355 | | 814 |
| OASIS | 24.3% | 405 | 84.6% | 368 | 89.4% | 306 | 97.7% | 21 | 83.3% | 492 | 67.9% | 380 |
| KISSME | 67.8% | 627 | 92.1% | 224 | 95.9% | 220 | 95.7% | 74 | 76.2% | 1750 | 40.8% | 7900 |
| RMML | 43.7% | 342 | 66.5% | 476 | 87.5% | 509 | 97.6% | 18 | 82.1% | 366 | M | |
| LMNN | 62.2% | 1680 | 88.4% | 261 | **97.6%** | 1850 | **97.8%** | 3634 | M | | M | |
| LRSML | 46.9% | 90 | 93.7% | 50 | 85.3% | 1093 | M | | M | | M | |
| FRML | 75.7% | 227 | **94.2%** | 391 | 97.0% | 371 | 90.5% | 48 | 76.1% | 172 | 69.1% | 931 |
| **FLRML** | **80.2%** | **37** | 93.5% | **14** | 96.3% | **10** | 95.9% | **7** | **83.4%** | **41** | **75.0%** | **101** |

Table 4: The classification accuracy (left) and training time (right, in seconds) of 4 metric learning methods without SVD preprocessing.

| datasets | NG20 | | RCV1 | | TDT2 | | MNIST | | M10-16 | | M40-16 | | M10-100 | | M40-100 | |
|---|---|---|---|---|---|---|---|---|---|---|---|---|---|---|---|---|
| SCML | M | | 93.9% | 8508 | **96.9%** | 211 | 91.3% | 310 | E | | E | | M | | M | |
| DRML | 25.1% | 2600 | 91.4% | 216 | 82.2% | 750 | 82.1% | 1109 | 72.5% | 6326 | M | | M | | M | |
| SGDINC | 54.0% | 3399 | **94.2%** | 1367 | **96.9%** | 1121 | **97.6%** | 44 | **83.5%** | 174 | **74.2%** | 163 | 56.9% | 4308 | 35.5% | 5705 |
| **M-FLRML** | **54.2%** | **26** | 92.7% | **11** | 95.0% | **14** | 96.1% | **1** | 83.0% | **2** | 74.0% | **2** | **82.7%** | **637** | **73.8%** | **654** |

handwritten characters dataset **MNIST** [37]; four voxel datasets of 3D models **M10-16**, **M10-100**, **M40-16**, and **M40-100** with different resolutions in $16^3$ and $100^3$ dimensions, respectively, generated from "ModelNet10" and "ModelNet40" [38] which are widely used in 3D shape understanding [39, 40, 41, 42, 43, 44, 45, 46, 47, 48, 49, 50]. To measure the similarity, the data vectors are normalized to the unit length. The dimensions $D$, the number of training samples $n$, the number of test samples $n_{test}$, and the number of categories $n_{cat}$ of all the datasets are listed in Table 1.

Different methods have different requirements for SVD preprocessing. In our experiments, a fast SVD algorithm [51] is adopted. The time $t_{svd}$ in SVD preprocessing is listed at the top of Table 3. Using the same decomposed matrices as input, seven methods are compared: three methods (LRSML, LMNN, and our FLRML) require SVD preprocessing; four methods (FRML, KISSME, RMML, OASIS) do not mention SVD preprocessing, but since they need to optimize large dense $\mathbb{R}^{D \times D}$ matrices, SVD has to be performed to prevent them from out-of-memory error on high-dimensional datasets. For all these methods, the rank for SVD is set as $r = \min(\text{rank}(\mathbf{X}), 3000)$. The rest four methods (SCML, DRML, SGDINC, and our M-FLRML) claim that there is no need for SVD preprocessing, which are compared using the original data matrices as input. Specifically, since the SVD calculation for datasets M10-100 and M40-100 has exceeded the memory limit of common PCs, only these four methods are tested on these two datasets.

Most tested methods use either pairwise or triplet constraints, except for LMNN and FRML that requires directly inputting the labels in the implemented codes. For the other methods, 5 triplets are randomly generated for each sample, which is also used as 5 positive pairs and 5 negative pairs for the methods using pairwise constraints. The accuracy is evaluated by a 5-NN classifier using the output metric of each method. For each low-rank metric learning method, the rank constraint for $\mathbf{M}$ is set as $d = 100$. All the experiments are performed on the Matlab R2015a platform on a PC with 3.60GHz processor and 16GB of physical memory. The code is available at https://github.com/highan911/FLRML.

## 4.2 Experimental Results

Table 3 and Table 4 list the classification accuracy (left) and training time (right, in seconds) of all the compared metric learning methods in all the datasets. The symbol "E" indicates that the objective

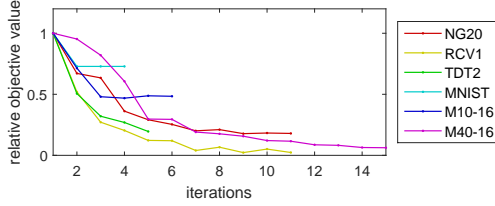
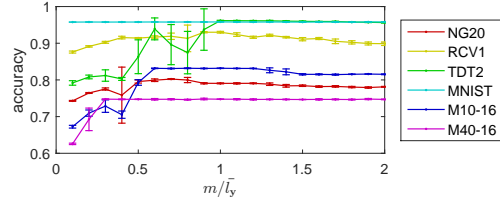

Figure 2: The convergence behavior of FLRML in optimization on 6 datasets.

Figure 3: The change in accuracy of FLRML with different $m/\bar{l}_\mathbf{y}$ values on 6 datasets.

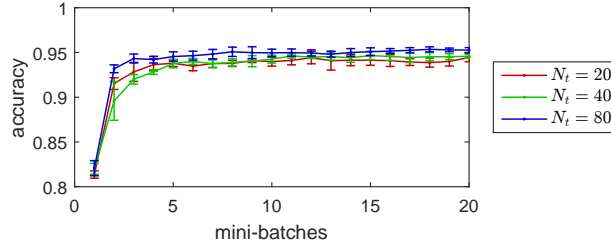

Figure 4: The change in accuracy of M-FLRML on "TDT2" with different $N_t$ values and number of mini-batches.

fails to converge to a finite non-zero solution, and "M" indicates that its computation was aborted due to out-of-memory error. The maximum accuracy and minimum time usage for each dataset are boldly emphasized.

Comparing the results with the analysis of complexity in Table 2, we find that for many tested methods, if the complexity or matrix is a polynomial of $D$, $n$ or $|\mathcal{T}|$, the efficiency on datasets with large numbers of samples is still limited. As shown in Table 3 and Table 4, FLRML and M-FLRML are faster than the state-of-the-art methods on all datasets. Our methods can achieve comparable accuracy with the state-of-the-art methods on all datasets, and obtain the highest accuracy on several datasets with both high dimensions and large numbers of samples.

Both our M-FLRML and SGDINC use mini-batches to improve efficiency. The theoretical complexity of these two methods is close, but in the experiment M-FLRML is faster. Generally, M-FLRML is less accurate than FLRML, but it significantly reduces the time and memory usage on large datasets. In the experiments, the largest dataset "M40-100" is with size $1,000,000 \times 118,116$. If there is a dense matrix of such size, it will take up 880 GB of memory. When using M-FLRML to process this data set, the recorded maximum memory usage of Matlab is only 6.20 GB (Matlab takes up 0.95 GB of memory on startup). The experiment shows that M-FLRML is suitable for metric learning of large-scale high-dimensional data on devices with limited memory.

In the experiments, we find the initialization of $\mathbf{s}$ usually converges within 3 iterations. The optimization on the Stiefel manifold usually converges in less than 15 iterations. Figure 2 shows the samples of convergence behavior of FLRML in optimization on each dataset. The plots are drawn in relative values, in which the values of first iteration are scaled to 1.

In FLRML, one parameter $m$ is about the margin in the margin loss. An experiment is performed to study the effect of the margin parameter $m$ on accuracy. Let $\bar{l}_\mathbf{y}$ be the mean of $\mathbf{y}_i^\top \mathbf{y}_i$ values, i.e. $\bar{l}_\mathbf{y} = \frac{1}{n}\sum_{i=1}^n (\mathbf{y}_i^\top \mathbf{y}_i)$. We test the change in accuracy of FLRML when the ratio $m/\bar{l}_\mathbf{y}$ varies between 0.1 and 2. The mean values and standard deviations of 5 repeated runs are plotted in Figure 3, which shows that FLRML works well on most datasets when $m/\bar{l}_\mathbf{y}$ is around 1. So we use $m/\bar{l}_\mathbf{y} = 1$ in the experiments in Table 3 and Table 4.

In M-FLRML, another parameter is the number of triplets $N_t$ used to generate a mini-batch. We test the effect of $N_t$ on the accuracy of M-FLRML with the increasing number of mini-batches. The mean values and standard deviations of 5 repeated runs are plotted in Figure 4, which shows that a larger $N_t$ makes the accuracy increase faster, and usually M-FLRML is able to get good results within 20 mini-batches. So in Table 4, all the results are obtained with $N_t = 80$ and $T = 20$.

# 5    Conclusion and Future Work

In this paper, FLRML and M-FLRML are proposed for efficient low-rank similarity metric learning on high-dimensional datasets with large numbers of samples. With a novel formulation, FLRML and M-FLRML can better employ low-rank constraints to further reduce the complexity and matrix size, based on which optimization is efficiently conducted on Stiefel manifold. This enables FLRML and M-FLRML to achieve good accuracy and faster speed on large numbers of high-dimensional data. One limitation of our current implementation of FLRML and M-FLRML is that the algorithm still runs on a single processor. Recently, there is a trend about distributed metric learning for big data [52, 53]. It is an interest of our future research to implement M-FLRML on distributed architecture for scaling to larger datasets.

**Acknowledgments**

This research is sponsored in part by the National Key R&D Program of China (No. 2018YF-B0505400, 2016QY07X1402), the National Science and Technology Major Project of China (No. 2016ZX 01038101), and the NSFC Program (No. 61527812).

## Footnotes

*Corresponding author: Yu-Shen Liu.

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
