[Reviews · NeurIPS 2019]

Reviewer 1



In this paper, the authors proposed Fast Low-Rank Metric Learning (FLRML) and M-FLRML for large-scale and high-dimensional metric learning problem, by employing the low rank constraints and the stochastic learning methods to reduce the computational complexity of the method. I think this paper is well prepared. I have the following comments. (1) In my opinion, it is a common way to process the large-scale and high dimensional data by using low-rank constraints and stochastic learning strategy for metric learning. Besides, the main idea of FLRML is to replace Y by $BV^T$ and is very similar with anchor-based strategy [1]. Please explain the difference between them and point out the main contribution of this paper. (2) Is there existing any theoretical results of the process of using $BV^T$ to replace Y, which ensures the performance of accelerated low-rank metric learning? (3) For the stochastic metric learning methods, there are some recent methods, which are not summarized in the related works of this paper, such as [2] and [3]. Meanwhile, what the differences between FLRML and OPML[3] and the method in [2]? [1] Liu, Wei, Junfeng He, and Shih-Fu Chang. "Large graph construction for scalable semi-supervised learning." In Proceedings of the 27th international conference on machine learning (ICML-10), pp. 679-686. 2010. [2] Qian, Qi, Rong Jin, Jinfeng Yi, Lijun Zhang, and Shenghuo Zhu. "Efficient distance metric learning by adaptive sampling and mini-batch stochastic gradient descent (SGD)." Machine Learning 99, no. 3 (2015): 353-372. [3] Li, Wenbin, Yang Gao, Lei Wang, Luping Zhou, Jing Huo, and Yinghuan Shi. "OPML: A one-pass closed-form solution for online metric learning." Pattern Recognition 75 (2018): 302-314.

Reviewer 2



Low-rank metric learning optimizes a metric matrix subject to low-rank constraints, preserving the intrinsic low-rank structure of the data. However, it still encounters scalability problem when handling large data. This work gives a new formulation that learns the low-rank cosine similarity metric by embedding the triplet constraints into a matrix to further reduce the complexity and the size of involved matrices. The idea of embedding the evaluation of loss functions into matrices is interesting. For Stiefel manifolds, rather than following the projection and retraction convention, it adopts the optimization algorithm proposed by Wen et al. (Ref. [3]). Generally, this paper is well-written with promising results. Here are some concerns: 1) The upper bound is set to 3000, which means the dimension $r$ is truncated regardless of the intrinsic value of very large and high-dimensional data. Is there any theoretical analysis or just an empirical value? 2) To make the gradient of $f(P)$ continuous, the smoothing function $\mu(x)$ is adopted for $max(0, x)$, what about the approximation loss between them? If not using such smoothing, how to optimize the problem in Eq. (6)? 3) The SVD pre-processing still needs expensive cost which cannot be avoided. It proposed mini-batch version which requires $O(D{n_I}^2 + Ddn_I d)$, i. e., calculating a descent direction from each mini-batch of data and updating the transformation matrix $L$ at a decreasing ratio. It would be nice to provide some theoretical analysis of this strategy. 4) Some parameter sensitivity examinations are expected, e.g., the rank constraint for $M$ is set to $d$=100, the number of triplet constraints, and the size of mini-batch. Minor issue: Line 93, the parameter "m" which is about the margin is undefined and some analysis is required here. ================ In the rebuttal, authors have well addressed most of the raised points. I vote for acceptance.

Reviewer 3



The paper is well written and structured. It tackles a relevant topic. The problem is well formulated, the method is exposed clearly and the simulation studies make a good case for the method.

[Author Response · NeurIPS 2019]

We thank the reviewers for their constructive comments and suggestions. We respond to each point individually.

**R1:** *More description to highlight the unique contribution.*
Compared with previous metric learning methods by using low-rank/online/stochastic strategies, that still encounter
the scalability problem when handling large data, our paper has two unique contributions as follows. (1) Our method
embeds the triplet constraints into a matrix, and further reduces the size of involved matrices by replacing $\mathbf{Y}$ with
$\mathbf{BV^T}$, which ensures the existence of the optimal solution in the reduced matrices. (2) By substituting the closed-form
optimal solution of $\mathbf{s}$, the optimization of positive semidefinite matrices is converted into the optimization on the Stiefel
manifold, which can be optimized more efficiently. These two contributions significantly reduce the complexity and the
size of involved matrices, which makes our method scalable to both high dimensions and large numbers of samples.

**R1:** *Explain the difference with some recent methods.*
We will add the missing references in "Related Work" and explain the difference with them in the revision. *Mini-SGD*
[Qian et al. ML 2015] uses the mini-batch strategy to optimize the metric matrix in the positive semidefinite cone.
Using the online metric learning strategy, *OPML* [Li et al. PR 2018] introduces a closed-form formula for updating the
metric matrix. However, the above two methods all require updating a large $D \times D$ matrix ($D$ is the dimensionality of
original data) at each iteration, and they can only learn from samples with hundreds of dimensions in their experiments.
In contrast, our method models the metric learning problem on the Stiefel manifold with much smaller size $r \times d$, which
significantly reduces the complexity and memory usage in optimization. In our experiments, the datasets can be with up
to one million dimensions (see Table 1 in our paper).

**R1:** *Theoretical results using $\mathbf{BV^T}$ to replace $\mathbf{Y}$ to ensure performance, and differences with anchor-based strategy.*
(1) If the linear equation system $\mathbf{Y} = \mathbf{LX}$ has a feasible solution of $\mathbf{L}$, it can always be transformed into a full-rank
linear equation system $\mathbf{YV} = \mathbf{LU\Sigma}$. According to Theorem 1 in our paper, since $\mathbf{YV} = \mathbf{B}$, all the possible solutions
can be covered by $\mathbf{B} \in \mathbb{R}^{d \times r}$, which ensures the performance of accelerated low-rank metric learning.
(2) For the anchor-based strategy in *AnchorGraph* [Liu et al. ICML 2010], the anchors are "local samples" with high
dimension $D$, and the algorithm represents each data point as a convex combination of its closest anchors. In contrast,
our method optimizes a smaller matrix $\mathbf{B} \in \mathbb{R}^{d \times r}$ to obtain the global optimal solution of a larger matrix $\mathbf{L}$, which is
different with the "local sampling" strategy used in *AnchorGraph*, as will be added in "Related Work" in the revision.

**R2:** *The upper bound of $r$.*
As a preprocessing parameter, the upper bound of $r$ is an empirical value. In general, a larger $r$ will increase memory
and computational costs, while a smaller $r$ may lose some intrinsic values. In our preliminary experiments under dataset
"TDT2", when $r$ varies in 500, 1000, 2000, 3000 and 4000, the accuracy just slightly changes to 0.948, 0.964, 0.965,
0.963 and 0.948, respectively. This shows that the variation of $r$ within a reasonable range merely affects computational
time and memory, but has little effect on accuracy. For fair comparison, we use the same $r$ for all compared methods.

**R2:** *What about the approximation loss of $\mu(x)$, and how to optimize the problem without using $\mu(x)$?*
The authors guess the approximation loss concerned by the reviewer is the difference between $\mu(x) = -\log(\sigma(-x))$
and $\max(0, x)$, where its maximum value is $-\log \frac{1}{2} \approx 0.69$ at the point $x = 0$. If not using $\mu(x)$, our model can still
be solved. However, due to the discontinuous gradient, the convergence requires more iterations, and the results are
less stable. For example, under the dataset "TDT2" in five repeated runs, when using $\mu(x)$, the average number of
iterations, average accuracy, and standard deviation are 6.6, 0.962, and 0.0008, respectively. In contrast, when only
using $\max(0, x)$, these values become 12.6, 0.955, and 0.012, respectively.

**R2:** *Theoretical analysis of stochastic strategy.*
The theoretical analysis of the stochastic strategy which updates $\mathbf{L}$ in step sizes by $1/\sqrt{I}$ decay can refer to the reference
[14] in our paper. We will add more description for [14] about the stochastic strategy in the revision.

**R2:** *Parameter sensitivity examinations.*
- The examination of different mini-batch sizes has been provided in Fig. 4, where $N_t$ is an indicator of mini-batch size.
- The examination of different $m$ has been provided in Fig. 3. We will explicitly add the definition of $m$ in the revision.
- $d$ is the target rank value in low-rank metric learning tasks, which is usually set according to user preference. We use
$d = 100$ in all experiments for a fair comparison, which is a moderate value so that most tested methods can achieve
good performance on most datasets, as mentioned in Section 4.1.
- Triplet sizes: 5 triplets are randomly generated for each sample (which is a moderate value for most datasets), and the
same triplet sets are used for all tested methods for a fair comparison, as mentioned in Section 4.1.

**R3:** *Potential applications and more explicit gaps on the previous literature.*
A more detailed highlight of contributions and differences with recent methods can refer to 1st Answer for Reviewer 1.
Metric learning has been widely used in various areas, such as dimensionality reduction, feature extraction, and
information retrieval. Our method can be applied to the scenario of learning metrics quickly on large numbers of
high-dimensional data with limited computing resources. We will introduce some application scenarios in the revision.

[Meta-Review · NeurIPS 2019]

The reviewers appreciated the computational improvements and the ideas (such as embedding the evaluation of cost into matrices) that went into them. Scores were fairly lukewarm before the rebuttal but the authors did a good job in the rebuttal to address all concerns.